# Efficient iron single-atom catalysts for selective ammoxidation of alcohols to nitriles

Kangkang Sun[1,2], Hongbin Shan[1], Helfried Neumann [2✉], Guo-Ping Lu [1✉] & Matthias Beller [2✉]

Zeolitic imidazolate frameworks derived $Fe_1$-N-C catalysts with isolated single iron atoms have been synthesized and applied for selective ammoxidation reactions. For the preparation of the different Fe-based materials, benzylamine as an additive proved to be essential to tune the morphology and size of ZIFs resulting in uniform and smaller particles, which allow stable atomically dispersed $Fe–N_4$ active sites. The optimal catalyst $Fe_1$-N-C achieves an efficient synthesis of various aryl, heterocyclic, allylic, and aliphatic nitriles from alcohols in water under very mild conditions. With its chemoselectivity, recyclability, high efficiency under mild conditions this new system complements the toolbox of catalysts for nitrile synthesis, which are important intermediates with many applications in life sciences and industry.

[1] School of Chemistry and Chemical Engineering, Nanjing University of Science & Technology, Xiaolingwei 200, Nanjing 210094, P. R. China. [2] Leibniz-Institute für Katalyse, Albert-Einstein-Str. 29a, 18059 Rostock, Germany. ✉email: Helfried.Neumann@catalysis.de; glu@njust.edu.cn; Matthias.Beller@catalysis.de

**N**itriles are core structures and intermediates of many pharmaceuticals, agrochemicals, and fine chemicals[1–4]. As an example, selected bio-active nitriles are shown Fig. 1a. Due to the fundamental importance and practical value of nitriles for many applications in our daily life, numerous protocols have been developed for their synthesis (Fig. 1b), including ammoxidation of toluenes[5–7], dehydration of aldoximes or amides[8,9], Sandmeyer reaction[10,11], Rosenmund von Braun reaction[12,13], cyanation of aryl halides, and oxidation of amines[14–18]. These methods commonly proceed at high temperature/pressure and/or employ stoichiometric amounts of toxic HCN or metal cyanides as starting materials, which also leads to large amounts of poisonous waste. In addition, for functionalized nitriles (see Fig. 1a), (chemo)selectivity can be a problem. Thus, exploring new more benign and green methodologies for the synthesis of functionalized nitriles continues to attract the interest of academic and industrial chemists[19–28].

Among the various recent methodology developments, ammoxidation of available aliphatic and benzylic alcohols with molecular oxygen (or air) as the sole oxidant is a green and sustainable route to access nitriles. In this respect, specifically the ammoxidation of bio-derived alcohols offers potential for renewable chemicals and building blocks[29,30]. Moreover, such methodology has high atom- and step-economy as water is formed as the only by-product in the whole process. Although a series of homogeneous catalysts have been developed, such as Cu/TEMPO, Fe/TEMPO[31–33], and these systems exhibit excellent catalytic activity, still catalyst recycling and product separation can be problematic. In this respect, heterogeneous catalysts offer advantages due to their superior recyclability, separation, and stability[34–36]. Thus, a series of precious metal-based heterogeneous catalysts such as $Ru(OH)_x/Al_2O_3$, Pt/GO, Ag/N-CS-1, and $Ru/MnO_2$-$r$ were reported for the ammoxidation of alcohols in the past decade (Supplementary Tables 1–2)[37–40].

However, intrinsic high cost, limited availability, and toxicity demand their replacement with more earth-abundant-metal catalysts. As an example, some of us reported the first ammoxidation of alcohols to nitriles with $Co_3O_4$-NGr/C or $Fe_2O_3$-NGr/C based

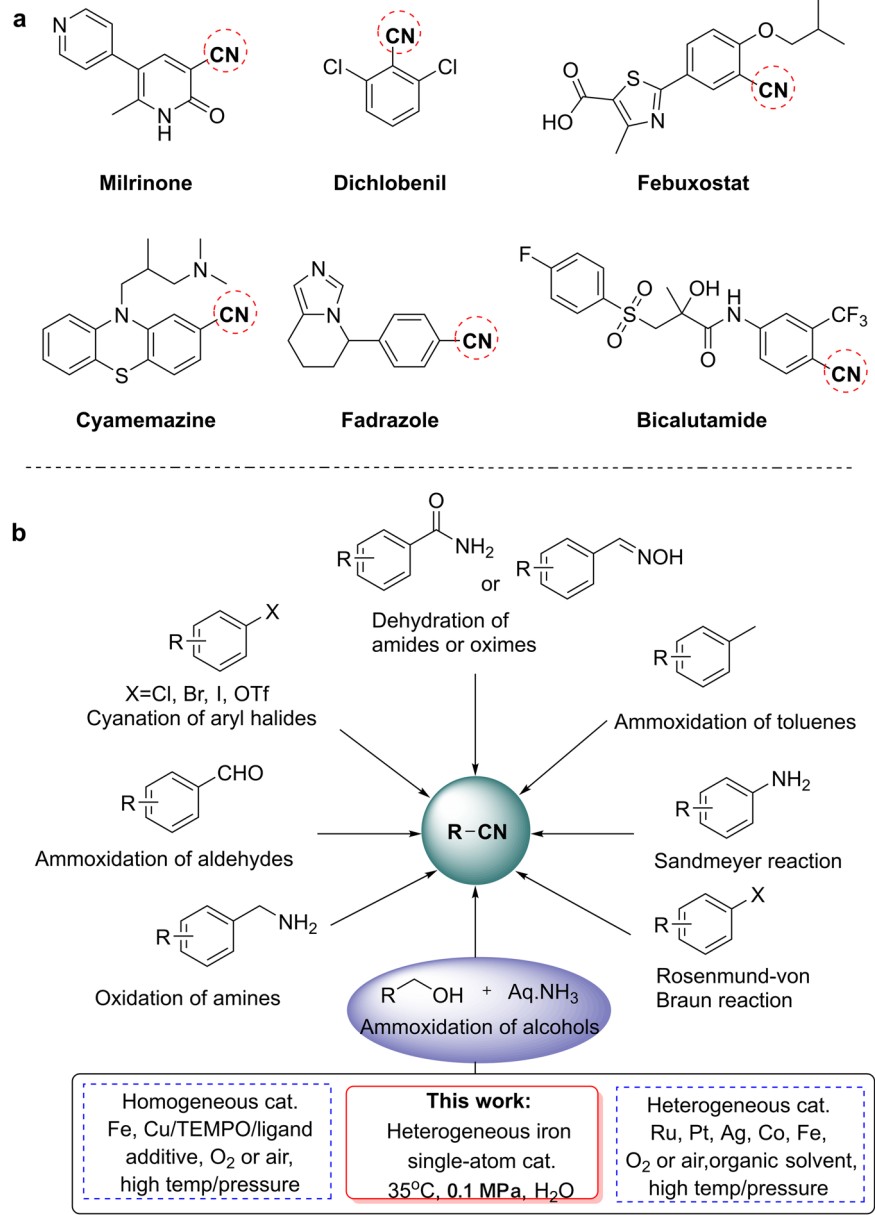

**Fig. 1 Concepts. a** Selected examples of current drugs and bio-active functionalized nitriles. **b** strategies for the synthesis of nitriles.

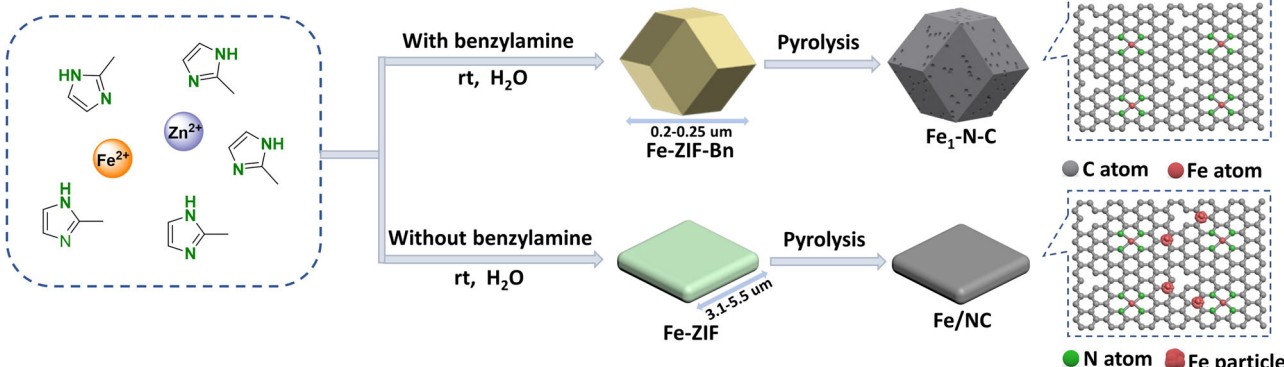

**Fig. 2 Catalyst preparation process.** Schematic illustration of the synthesis of Fe₁-N-C catalyst.

heterogeneous catalysts[41]. Notably, high-temperature, pressure and organic solvents were required even for activated substrates[42]. Later on, Gao and co-workers developed well-designed meso-Co–N/C(800) and meso-N/C-900 catalysts for the synthesis of nitriles from alcohols[43,44]. Although these catalysts exhibit excellent activity, comparably high temperature and pressure as well as the use of special polypyridyl ligands were still inevitable. To overcome these limitations and to improve the activity of the currently known heterogeneous catalyst systems, we became interested to prepare materials with highly distributed metal atoms or ideally single metal sites on the respective support. Among the various metals, which can be used for single-atom catalysts (SACs) preparation, iron is particularly interesting due to its wide redox chemistry, wide availability, low costs, and low toxicity[45].

Zeolitic imidazolate frameworks (ZIFs) with large surface area, high pore volume, and rich-N content constitute popular precursors for the preparation of many heterogeneous catalysts[46–49]. However, most of the reported methods for synthesizing iron-doped ZIFs make use of organic solvents. The aqueous synthesis of such materials, e.g., ZIF-8 is not trivial as it can generate by-products or amorphous compounds, and the synthesis generally requires a higher reaction temperature (60 °C or 120 °C) or a longer reaction time (24 h or 48 h). Moreover, the agglomeration of metals during the calcination process will reduce the atom utilization efficiency[50–53]. This so-called 'coordination modulation strategy' has been reported in the literature for controlling the MOF particle size, morphology, and nucleation rate specifically using different organic amines such as amino acids[54], triethylamine[55], and n-butylamine[56]. These organic amines act as crystallization modulators and can compete with bridging ligands to regulate the crystallization process[57]. Bearing these in mind, herein we report new catalysts with highly distributed iron atoms supported on nitrogen-doped porous carbon derived from benzylamine-modified ZIFs. The new synthesis of ZIF precursors not only accelerates the nucleation rate of ZIFs in water and allows to obtain more uniform and smaller size ZIFs particles, but also protects the iron atoms from significant aggregation during calcination. The obtained catalyst (hereafter abbreviated as Fe₁-N-C) has a relatively high loading (2.2 wt.%) of highly dispersed iron atoms and shows excellent performance for the preparation of diverse nitriles from easily available alcohols and aqueous ammonia under mild conditions.

## Results

**Catalyst synthesis.** The preparation of the supported Fe₁-N-C catalyst is shown in Fig. 2. In brief, 2-methylimidazole and benzylamine (1:1) were dissolved in water. This mixture was added to the solution of iron(II) sulfate heptahydrate and zinc nitrate hexahydrate under stirring at room temperature for 4 h. The resulting solid was collected by centrifuge, washed with methanol,

and dried under vacuum. The resulting powder (Fe-ZIF-Bn) was pyrolyzed at 900 °C under a constant argon flow for 2 h to afford Fe₁-N-C. During this process, Zn species evaporate, which increases the specific surface area and number of defects of the resulting carbon material[58,59]. At the same time, Zn as a self-sacrificial template can effectively eliminate the agglomeration of iron nanoparticles, which is beneficial for the formation of iron single atoms. In control experiments, Fe-ZIFs particles with different ratios of benzylamine/2-MeIm (Fe-ZIF-BnX, X denotes the mole ratio of benzylamine/2-MeIm) and different additives were also synthesized (Fe-ZIF-TEA and Fe-ZIF-NaOH).

**Catalyst activity tests.** To evaluate the catalytic performance of the Fe₁-N-C catalyst for the ammoxidation of alcohols to nitriles, benzyl alcohol (**1a**) was chosen as a well-known benchmark substrate. Typically, catalytic reactions were carried out in water at 35 °C with aqueous ammonia as the nitrogen source at atmospheric pressure of air as "green" oxidant (Table 1). Obviously, blank experiments (without any Fe catalyst) did not work (Table 1, entries 1–3). Comparing the performance of Fe/NC (derived from Fe-ZIF without benzylamine modification) with Fe₁-N-C a significantly improved yield of benzonitrile **3a** is observed, which reached 99% after prolonging reaction time to 24 h (Table 1, entries 4-7). Notably, among the six Fe-N-C nanocomposites derived from Fe-ZIFs particles with different ratios of benzylamine/2-MeIm and different additives, Fe₁-N-C exhibits the best catalytic performance (Supplementary Table 3). Commercially available ZnO, Zn(NO₃)₂.6H₂O and FeSO₄.7H₂O were also applied for control experiments and none of them showed any activity and formation of benzonitrile (Table 1, entries 8-10). For comparison, we evaluated the performance of Fe₂O₃-NGr/C[42] and a series of other iron-based heterogeneous catalysts (Table 1, entries 11–14), but none of them showed significant activity under these mild reaction conditions.

**Catalyst characterization.** To understand the specific behavior of the Fe₁-N-C catalyst and its structural features, detailed characterization experiments were conducted. Unlike the simple Fe-ZIFs, the powder X-ray diffraction (PXRD) patterns of the Fe-ZIF-Bn match well with the simulated ZIF-8 material, demonstrating the improved crystallinity of the Fe-ZIF-Bn material (Supplementary Fig. 1a). More specifically, Fe-ZIF-Bn presents a distinct dodecagon structure with small particle size of around 200 nm (Fig. 3a). Scanning electron microscopy (SEM) revealed the morphological differences of Fe-ZIFs particles prepared with different additives and different ratios of benzylamine to 2-MeIm (Supplementary Fig. 2-3). As the amount of benzylamine increases, the morphology of Fe-ZIFs precursor becomes more regular and the particle size becomes smaller. In contrast, the

addition of triethylamine and NaOH had only little effect on the morphology of Fe-ZIFs. Apparently, the basic nature of the additives is not the only factor affecting the crystal structure and morphology of Fe-ZIFs (Supplementary Figs. 1–3).

**Table 1 Catalyst evaluation for the ammoxidation of benzyl alcohol[a].**

$$1a \quad + \quad aq.\ NH_3\ H_2O \quad \xrightarrow[H_2O]{Catalyst} \quad 3a$$

| Entry | Catalyst | Time (h) | Conv. (%) | Yield (%)[b] |
|---|---|---|---|---|
| 1 | / | 24 | 0 | 0 |
| 2 | NC[c] | 24 | 0 | 0 |
| 3 | N-C[d] | 24 | 15 | 0 |
| 4 | Fe/NC[e] | 12 | 48 | 47 |
| 5 | Fe/NC[e] | 24 | 71 | 69 |
| 6 | Fe$_1$-N-C | 12 | 66 | 66 |
| **7** | **Fe$_1$-N-C** | **24** | **>99** | **>99** |
| 8 | ZnO | 24 | 0 | 0 |
| 9 | Zn(NO$_3$)$_2$.6H$_2$O | 24 | 0 | 0 |
| 10 | FeSO$_4$.7H$_2$O | 24 | 0 | 0 |
| 11 | Fe-NC[f] | 24 | 18 | 3 |
| 12 | Fe/AC[f] | 24 | <5 | <5 |
| 13 | Fe/SiO$_2$[f] | 24 | <5 | <5 |
| 14 | Fe$_2$O$_3$-NGr/C | 24 | 13 | 12 |

The most active catalyst is shown in bold.
[a]Reaction conditions: **1a** (0.2 mmol), 20 mg catalyst (3.9 mol% Fe), aq. NH$_3$·H$_2$O (150 mg, 25–28 wt%), 35 °C, 0.1 MPa air, 1.5 mL H$_2$O.
[b]Conversion and yield were determined by GC analysis using hexadecane as an internal standard.
[c]The NC material was obtained from the carbonation of ZIF-8.
[d]The N-C material was obtained from the carbonation of benzylamine-modified ZIF-8.
[e]Fe/NC derived from Fe-ZIFs without benzylamine modification.
[f]These catalysts were prepared by typical impregnation method.

After pyrolysis at 900 °C, the morphology of the Fe-ZIF-Bn particles remained basically unchanged, and no obvious aggregation of Fe atoms was found (Fig. 3b). Notably, individual iron atoms (marked by yellow circles) anchored on the entire carbon architectures are clearly observed by HAADF-STEM (Fig. 3c). Meanwhile, energy-dispersive X-ray spectroscopy (EDX) mappings of Fe$_1$-N-C show the homogeneous distribution of Fe, C, and N species (Fig. 3d). This result is in good agreement with powder XRD curves, which show only the broad shoulder assigned to graphite (002) and (100) plane (Supplementary Fig. 1c)[60].

The original Fe-ZIF (without benzylamine modification) exhibit the morphology of nanosheets with 3100–3700 nm length and 430–470 nm thickness. After pyrolysis, this structure of Fe-ZIF was severely destroyed (Supplementary Fig. 2) and iron nanoparticles were formed according to XRD and TEM results (Supplementary Fig. 1b). Thus, the simple addition of benzylamine plays a key role in the formation of the single metal sites. Elemental analysis of Fe$_1$-N-C and Fe/NC show that the active catalyst material has a higher nitrogen content probably due to the use of benzylamine (Supplementary Table 5).

After pyrolysis at high temperature, the morphology of Fe-ZIF-Bn was not severely damaged owing to its excellent crystallinity and stability. Therefore, compared with Fe/NC, Fe$_1$-N-C has higher BET surface area (Fe$_1$-N-C: 817 m$^2$/g, Fe/NC: 712 m$^2$/g), pore volume (Fe$_1$-N-C: 0.84 cm$^3$/g; Fe/NC: 0.30 m$^3$/g) (Supplementary Fig. 4), and increased N content (Fe$_1$-N-C: 4.63%; Fe/NC: 3.96%). The high specific surface area, pore volume, and the co-existence of mesopores and micropores should be advantageous for the absorption and transport of air and reactants in the ammoxidation reaction, thereby contributing to the excellent catalytic activity[61]. According to SEM, XRD, BET and literature results[62–64], benzylamine as a crystallization regulator proved to be essential to tune the morphology and size of ZIFs in water resulting in uniform and small particles, while also accelerating the nucleation speed and increasing the crystallinity, which seems favorable for the formation of stable atomic-level dispersion Fe-N$_4$ active sites.

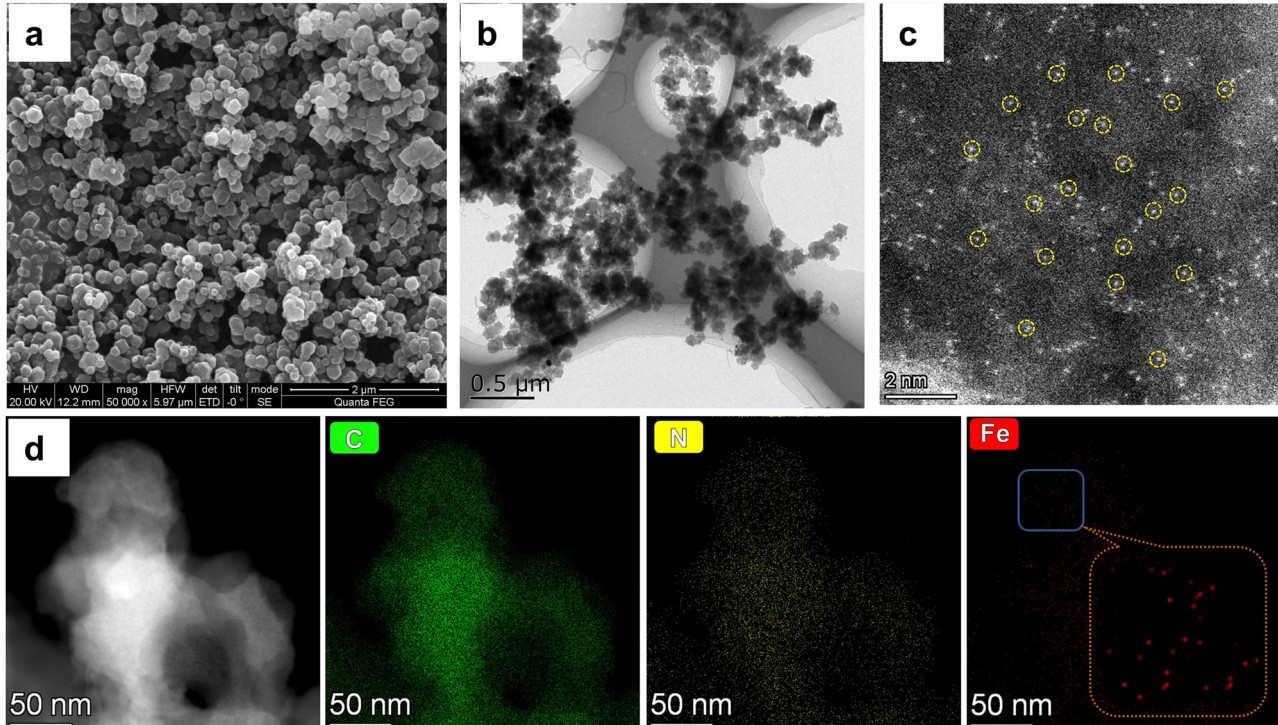

**Fig. 3 Representative electron microscopy images. a** SEM, **b** TEM, **c** HAADFSTEM, **d** EDS mapping images (C, N, and Fe) of Fe$_1$-N-C.

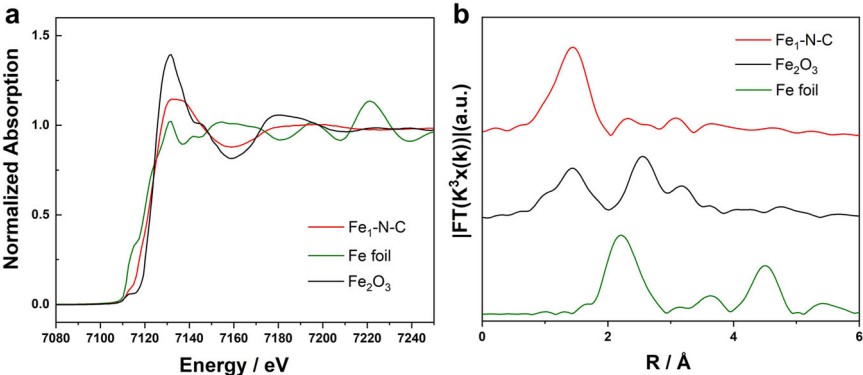

**Fig. 4 Structural characterizations of Fe₁-N-C. a** XANES spectra and **b** FT EXAFS spectra of Fe foil, Fe₂O₃, and Fe₁-N-C at the Fe K-edge.

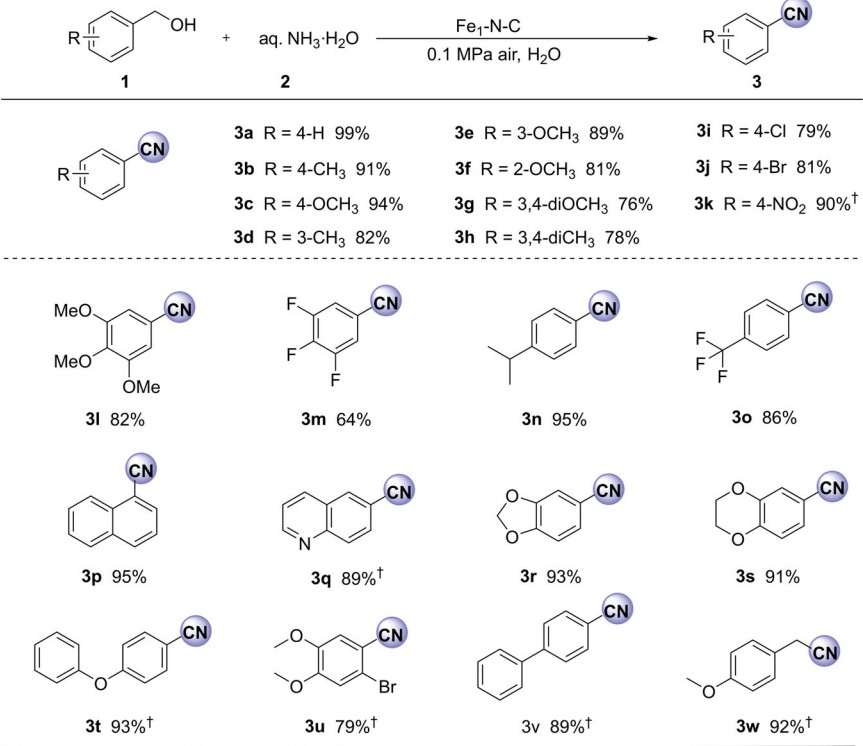

**Fig. 5 Ammoxidation of alcohols catalyzed by Fe₁-N-C.** Reaction conditions: alcohol (0.2 mmol), 20 mg catalyst (3.9 mol% Fe), aq. NH₃·H₂O (150 mg, 25–28 wt%), 35 °C, 0.1 MPa air, 1.5 mL H₂O, isolated yields. †60 °C.

To further elucidate the bonding state and local coordination structure of Fe atoms of Fe₁-N-C, Fe K-edge synchrotron radiation-based X-ray absorption near-edge structure (XANES) measurements were performed with Fe foil and Fe₂O₃ as references. The near-edge feature of Fe₁-N-C is located between Fe foil and Fe₂O₃ (Fig. 4a), indicating the electronic structure of $Fe^{\delta+}$ ($0 < \delta < 3$), which coincides with the XPS results. At the same time, there is a small peak at 7113 eV, which usually originates from the 1 s → 4pz transition and the charge transfer between the ligand and the metal, implying the square-planar Fe-N₄ structure. The Fourier-transformed k3-weighted extended X-ray absorption fine structure (FTEXAFS) spectrum in the R space (Fig. 4b) shows a peak at 1.45 Å, which can be assigned to Fe–N scattering path, and no typical Fe−Fe (2.15 Å) peaks can be detected, confirming the sole presence of isolated metal sites in Fe₁-N-C. EXAFS fitting analysis (Supplementary Fig. 5) results in a coordination number of Fe–N of approximately 4.3 with a bond distance of 1.97 Å (Supplementary Table 4). Again, these results confirm the proposed Fe–N₄ moieties in Fe₁-N-C[60,65].

Finally, X-ray photoemission spectroscopy (XPS) of Fe₁-N-C was used to determine the surface elemental compositions and electronic state. Fe, C, N, O are clearly identified (Supplementary Fig. 6). Fe 2p spectrum of Fe₁-N-C and Fe/NC displayed two peaks centered at the binding energies of 724.2 and 710.4 eV, assigned to Fe $2p_{1/2}$ and Fe $2p_{3/2}$, respectively. (Supplementary Fig. 7)[66–68]. The XPS spectrum for N 1s displays four peaks with binding energies of 401.3, 400.7, 399.8, and 398.4 eV, which are ascribed to oxidized N, graphitic N, pyrrolic N, and pyridinic N, respectively (Supplementary Fig. 7b)[69,70]. In addition, the Raman spectrum presented in Supplementary Fig. 8 shows that there are more defects in Fe₁-N-C due to the higher nitrogen content[71].

**Substrate scope.** After obtaining optimal conditions for the benchmark reaction, we explored the generality of this active catalyst for the synthesis of various nitriles. As shown in Fig. 5, diverse benzyl alcohols bearing electron-donating or -withdrawing groups are selectively oxidized to the corresponding nitriles with good to very

- **Heterocyclic nitriles, allylic nitriles, and aliphatic nitriles**

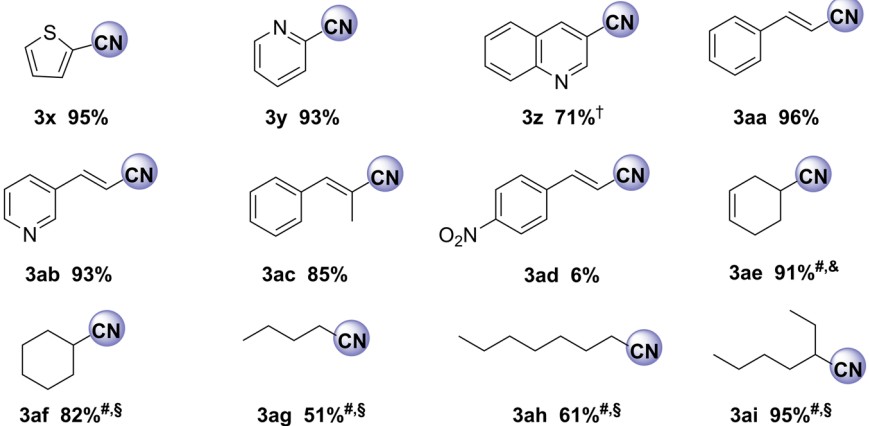

**Fig. 6 Synthesis of heterocyclic, allylic, and aliphatic nitriles.** Reaction conditions: 0.2 mmol alcohol, 20 mg catalyst (3.9 mol% Fe), aq. $NH_3 \cdot H_2O$ (150 mg), 35 °C, 0.1 MPa air, 1.5 mL $H_2O$, isolated yields; #Yields were determined by GC using n-hexadecane as internal standard; †60 °C; §130 °C, 2 MPa air; &140 °C, 3 MPa air.

**Fig. 7 Nitriles produced from renewable aliphatic alcohols.** Reaction conditions: 0.2 mmol alcohol, 20 mg catalyst (3.9 mol% Fe), aq. $NH_3 \cdot H_2O$ (150 mg, 25–28 wt%), 130 °C, 2 MPa air, 1.5 mL $H_2O$. Yields were determined by GC using n-hexadecane as internal standard.

good yields (Fig. 5, **3a–3f**, **3i**, **3j**, **3n** and **3o**). Except for 4-nitrobenzyl alcohol (**3k**), all reactions took place at 35 °C and 0.1 MPa of air. Similarly, the ammoxidation of di- and tri-substituted benzylic alcohols and a naphthalene-based substrate provided the desired products with good efficiency (**3g**, **3h**, **3l**, **3m**, **3u**, **3p**). It is noteworthy that more demanding substrates, such as quinolin-5-ylmethanol, piperonol, (2,3-dihydro-benzo[1,4]dioxin-6-yl)-methanol, (4-phenoxyphenyl)methanol, 2-bromo-4,5-dimethoxybenzyl alcohol, and biphenyl-4-yl methanol, are selectively converted to corresponding nitriles in good yields (**3q–3v**).

Next, a series of more challenging substrates including heterocyclic, allylic, and aliphatic alcohols were employed (Fig. 6). Here, heterocyclic nitriles and allyl nitriles were successfully synthesized by this protocol with excellent yields (**3x–3ac**). Compared with benzylic alcohols, aliphatic ones usually have significantly lower reactivity. Indeed, most of the reported catalyst systems for benzylic alcohol oxidations are ineffective in the ammoxidation of aliphatic primary alcohols[32,72,73]. In contrast, utilizing our $Fe_1$-N-C as the catalyst, aliphatic alcohols produced the desired products on moderate to excellent yields (**3ae–3ai**, 51-95%), albeit the reaction temperature and the pressure of air had to be increased. In addition, when 1-octanol, 2-ethylhexyl alcohol, and cyclohexylmethanol were used as the substrate, the catalytic activity of $Fe_2O_3$-NGr/C[42] is much lower than that of $Fe_1$-N-C (Supplementary Table 2). Looking at the successful conversion of aliphatic alcohols to nitriles, the selective formation of **3ae** is especially remarkable as the C-C double bond is not touched in this process. This observation inspired us to explore the transformation of four different terpene-based alcohols, e.g., perillyl alcohol, myrtenol, citronellol, and geraniol, which are available from biomass[74–76]. In this context, it should be noted

that considerable attention has been given to valorize renewable biomass resources into fine chemicals in recent years[77,78]. As shown in Fig. 7, the desired renewable nitriles are obtained highly selectively.

**Gram scale reactions.** To demonstrate the utility of this catalytic system, $Fe_1$-N-C was used for the gram-scale synthesis of various nitriles. Despite the lower oxygen content and the limitation of oxygen transfer from gas phase to liquid phase, for all the tested substrates high yields were obtained on multi g-scale also with lower catalyst loading (Supplementary Fig. 9).

**Reaction mechanism.** Several control experiments were performed to investigate the reaction pathway for the synthesis of nitriles. First, the reaction was carried out in the absence of aqueous ammonia under optimal conditions, and benzaldehyde was obtained with 46% yield, thus indicating that aqueous ammonia also promotes the alcohol oxidation process. Then, we shortened the reaction time to 4 hours and found that the yield of benzonitrile obtained with benzaldehyde as the starting material was much higher than that obtained with benzyl alcohol as the starting material. In addition, in the model reaction benzaldehyde is detected by GM-MS in the early stage of the reaction. These results indicated that benzaldehyde is the first intermediate of this reaction sequence. Moreover, the catalytic oxidation of benzyl alcohol to benzaldehyde is the rate-limiting step in the formation of nitriles. Based on the above-described results and related published research[42,43], a possible mechanism of the reaction is proposed as shown in Supplementary Fig 10.

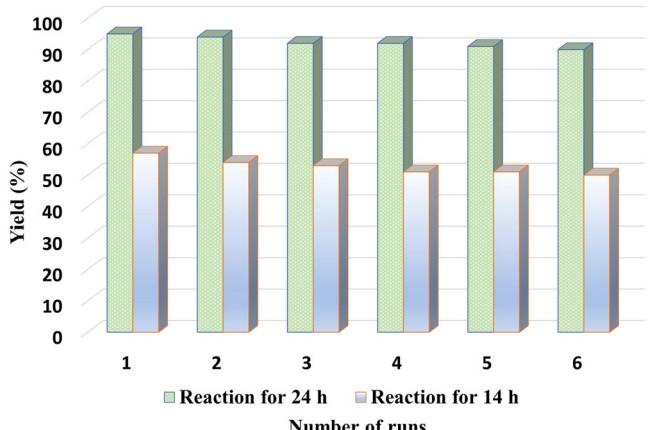

**Fig. 8 Recyclability of Fe₁-N-C for the synthesis of benzonitrile.** Reaction conditions: 20 mmol benzyl alcohol, 1 g catalyst, 150 mg of aq. $NH_3 \cdot H_2O$ (25–28 wt%) for each 0.2 mmol substrate, 0.1 MPa oxygen, 100 mL $H_2O$, 70 °C, 24 h and 14 h. Yields were determined by GC using n-hexadecane as standard.

**Catalyst recycling.** The recycling and reusability of heterogeneous catalyst are essential for sustainable and practical applications. In this respect, we tested the reusability of Fe₁-N-C for the model reaction using gram scale reactions conditions. As shown in Fig. 8, the catalytic activity and selectivity of Fe₁-N-C remained high after six runs and the change in the product yield was negligible, demonstrating the high durability of this catalyst. Another important aspect to be evaluated is the leaching of iron in the solution. Therefore, a hot-filtration experiment was performed, and the results are given in Supplementary Fig. 11. After filtering the catalyst, no further increase in yield was observed. Moreover, the reaction restarted after adding the removed catalyst. In agreement with this observation, no leaching of iron was detected in the solution after reaction by ICP analysis, which confirmed the excellent stability of the catalyst (Supplementary Table 6).

In conclusion, a convenient and scalable strategy is presented for the synthesis of efficient iron catalysts on N-doped carbon via pyrolysis of benzylamine-modified ZIFs. This not only allows to prepare catalysts with highly distributed single iron sites but also with comparable high metal loading. The presented catalyst displays excellent activity for the ammoxidation of various benzylic, heterocyclic, allylic, and aliphatic alcohols to give the desired nitriles. Moreover, the catalyst can be easily recycled and reused without significant loss in activity. In general, we believe the combination of this iron single-atom catalyst, atmospheric air, alcohols, and aqueous ammonia as starting materials make this process an ideal example for a green and more sustainable synthesis of nitriles.

## Methods

**Materials and general analytical methods.** All chemicals such as 2-methylimidazole (2-MeIm), $FeSO_4 \cdot 7H_2O$, $Zn(NO_3)_2 \cdot 6H_2O$, benzylamine, benzyl alcohol, and aqueous ammonia (25–28 wt%) were purchased from commercial suppliers and used directly without further purification.

[1]H NMR and [13]C NMR spectra were recorded on Bruker 300/400 and Bruker AV500M spectromete. GC-MS was performed on an ISQ Trace 1300 in the electron ionization (EI) mode. GC analyses are performed on an Agilent 7890A instrument (Column: Agilent 19091J-413: 30 m × 320 μm × 0.25 μm, carrier gas: $H_2$, FID detection. XRD analysis was performed on a Shimadzu X-ray diffractometer (XRD-6000) with Cu Kα irradiation. Transmission electron microscopy (TEM) images were taken using a PHILIPS Tecnai 12 microscope operating at 120 kv. Scanning electron microscopy (SEM) images were performed using a Hitachi S-4800 apparatus on a sample powder previously dried and sputter-coated with a thin layer of gold. Atomic-resolution HAADF-STEM images and EDS mappings were taken using a FEI Titan Cubed Themis G2 300S/TEM with a probe corrector and a

monochromator at 200 kV. X-ray photoelectron spectroscopy (XPS) was performed on an ESCALAB 250Xi spectrometer, using an Al Kα X-ray source (1350 eV of photons) and calibrated by setting the C 1s peak to 284.80 eV. Inductively coupled plasma mass spectrometry (ICP-OES) was analyzed on Varian (Agilent) 715-ES ICP-Emission-Spectrometer. BET surface area and pore size measurements were performed with $N_2$ adsorption/desorption isotherms at 77 K on a Micromeritics ASAP 2020 instrument. Before measurements, the samples were degassed at 150 °C for 12 h. Elemental analysis were performed on VARIO EL CUBE. The X-ray absorption spectra (XAS) including X-ray absorption near-edge structure (XANES) and extended X-ray absorption fine structure (EXAFS) of the samples at Fe K-edge (7712 eV) were collected at the Singapore Synchrotron Light Source (SSLS) center, where a pair of channel-cut Si (111) crystals was used in the monochromator. The Fe K-edge XANES date were recorded in a transmission mode. Fe foil, and $Fe_2O_3$ were used as references. The storage ring was working at the energy of 2.5 GeV with an average electron current of below 200 mA. The acquired EXAFS date was extracted and processed according to the standard procedures using the ATHENA module implemented in the IFEFFIT software packages.

**Synthesis of ZIF-8-Bn and ZIF-8.** ZIF-8-Bn was synthesized with the following steps. Typically, 8 mmol $Zn(NO_3)_2 \cdot 6H_2O$ was added to 80 mL $H_2O$ under ultrasonic. Meanwhile, 32 mmol 2-MeIm and 32 mmol benzylamine were added to 80 mL $H_2O$ by sonication for 10 min. The obtained salt solution was added to the above 2-MeIm solution under vigorous stirring at room temperature for 4 h. The obtained sample was centrifuged and washed with methanol. The final product was dried under vacuum at 80 °C for 12 h (Yield: 2.01 g). The same procedure was used for the synthesis of ZIF-8 except that no benzylamine was used (Yield: 1.70 g).

**Synthesis of Fe-ZIF-Bn, Fe-ZIF-TEA, Fe-ZIF-NaOH, Fe-ZIF-Bn8, Fe-ZIF-Bn4, Fe-ZIF-Bn2, and Fe-ZIF.** Fe-ZIF-Bn was synthesized with the following steps. Typically, 32 mmol 2-MeIm and 32 mmol benzylamine were added to 80 mL $H_2O$ under ultrasonic. Then, 0.38 mmol $FeSO_4 \cdot 7H_2O$ and 7.62 mmol $Zn(NO_3)_2 \cdot 6H_2O$ were dissolved in 80 mL $H_2O$ and added to the above mixture under vigorous stirring at room temperature for 4 h. Finally, the Fe-ZIF-Bn was separated, washed (methanol) and dried at 80 °C under vacuum (Yield: 2.05 g).

Use triethylamine (TEA) and NaOH instead of benzylamine to synthesize Fe-ZIF-TEA and Fe-ZIF-NaOH, respectively. The same procedure was used for the synthesis of Fe-ZIF-Bn8, Fe-ZIF-Bn4, Fe-ZIF-Bn2 and Fe-ZIF except that 4 mmol, 8 mmol, 16 mmol, and no benzylamine was used, respectively.

**Synthesis of Fe₁-N-C, Fe-N-C1, Fe-N-C2, Fe-N-C3, Fe-N-C4, Fe-N-C5, Fe/NC, NC, and N-C.** The pyrolysis of Fe-ZIF-Bn was carried out in a tubular furnace under a nitrogen atmosphere. The sample was heated from room temperature to 900 °C at a heating rate of 5 °C/min and kept at this temperature for 2 h. The resulting catalyst was denoted as Fe₁-N-C (Yield: 503 mg). The Fe-N-C1, Fe-N-C2, Fe-N-C3, Fe-N-C4, Fe-N-C5, Fe/NC, NC and N-C were prepared by a similar method using Fe-ZIF-Bn8, Fe-ZIF-Bn4, Fe-ZIF-Bn2, Fe-ZIF-TEA, Fe-ZIF-NaOH, Fe-ZIF, ZIF-8, and ZIF-8-Bn instead of Fe-ZIF-Bn, respectively.

**Synthesis of Fe-NC, Fe/AC, and Fe/SiO₂.** Fe-NC, Fe/AC, and Fe/SiO₂ were prepared by the typical impregnation method as follows: 1.0 g support (NC (derived from ZIF-8), activated carbon (AC), and $SiO_2$) was dispersed into 30 mL aqueous solution of metal precursors ($FeSO_4 \cdot 7H_2O$, 110 mg) under ultrasonic. Lysine aqueous solution (0.53 M, 5 mL) was then added into the mixture with vigorous stirring for 30 min. To this suspension, $NaBH_4$ aqueous solution (0.05 M, 25 mL) was added dropwise, the mixture was further stirred for 60 min and then aged for 24 h. Finally, the solid was separated, washed (water and ethanol) and dried under vacuum.

**Catalytic reactions.** Typically, a calculated amount of Fe₁-N-C (20 mg, 3.9 mol% of Fe) and 150 mg aq. $NH_3$ (25–28 wt%) in $H_2O$ (1 mL) were placed in a 25 mL sealed tube, and primary alcohol (0.2 mmol) was added to the mixture under atmospheric air with a magnetic stirrer to initiate the reaction at 35 °C for 24 h. After reaction, the mixture was extracted with ethyl acetate and the catalyst was isolated from the solution by centrifugation and directly reused after washing and drying. The product was analyzed by GC/MS (ISQ Trace 1300). The conversion and yield were determined by GC (Agilent 7890A) with hexadecane as an internal standard. The corresponding nitrile was purified by column chromatography.

**Gram scale reactions and catalyst recycling.** (a) The magnetic stir bar, $H_2O$ (100 mL), aq. $NH_3 \cdot H_2O$ (25–28 wt%,150 mg for each 0.2 mmol substrate), and corresponding substrates (20 mmol) were placed in a 300 mL round bottom flask. Then, the Fe₁-N-C (1 g) was added to the mixture under atmospheric oxygen with a magnetic stirrer to initiate the reaction at 70 °C for 24 h. After reaction, the mixture was extracted with ethyl acetate and the catalyst was isolated from the solution by centrifugation. The yield of the product was determined by GC (Agilent 7890 A) with hexadecane as an internal standard. (b) To a Teflon or glass fitted 300 mL autoclave, the magnetic stirring bar, and corresponding substrates (20 mmol) were transferred and then 100 mL of $H_2O$ was added. Then after, the

required amount of catalyst (Fe$_1$-N-C; 250 mg) was added. The autoclave was pressurized with 20 bar air. The autoclave was placed into an aluminum block preheated at 100 °C and the reaction was stirred for 10 h. After completion of the reaction, the autoclave was cooled to room temperature. The remaining air was discharged, and the products were removed from the autoclave. The mixture was extracted with ethyl acetate and the catalyst was isolated from the solution by centrifugation. The yield of the product was determined by GC (Agilent 7890 A) with hexadecane as an internal standard. After each cycle, the catalyst was isolated from the solution by centrifugation, washed three times with methanol, and dried under vacuum to remove the residual solvent and then reused for another reaction cycle.

## Data availability

The data that support the findings of this work are available from the corresponding author upon reasonable request.

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

## Acknowledgements
We gratefully acknowledge the Fundamental Research Funds for the Central Universities (30920021120), the National Natural Science Foundation of China (32001266) for financial support. We also acknowledge financial and general support from the European Research Council (EU project 670986-NoNaCat) and the State of Mecklenburg-Vorpommern.

## Author contributions
M.B., G.L., and K.S. conceived and designed the experiments. K.S. and H.S. performed the experiments and analyzed the data. H.N. participated in the discussions and supported the project. M.B., G.L., and K.S. prepared the manuscript with feedback from all authors.

## Funding

## Competing interests
The authors declare no competing interests.
