## [Peer Review File · Nature Communications]

Title: Efficient iron single-atom catalysts for selective ammoxidation of alcohols to nitrilesREVIEWER COMMENTS

Reviewer #1 (Remarks to the Author):

Sun et al. reported the ammoxidation of alcohols to nitriles using the Fe-N on carbon materials derived from ZIFs. Compared with the previous work, a significant breakthrough is the catalyst could work mild reaction temperatures and pressures. The results are interesting and might be suitable for publication in Nat Commun. However, how could the catalyst work to outperform the previous ones is still not fully discussed. I recommended publication of the manuscript after a revision.

(1) The reaction process involving multiple reaction steps should be discussed. Generally, the dehydrogenation of alcohol to aldehyde/ketone is a rate-control step (RDS). How about the RDS of this Fe-NC catalyzed reaction? Will the Fe-NC could catalyze the alcohol dehydrogenation under the given reaction conditions?

(2) Attention should be focused on the stability of such Fe-NC catalyst. In order to confirm its heterogeneous feature, the classical hot-filtration experiment and ICP analysis of the reaction liquor should be performed.

(3) Comparison of the Fe-NC catalyst with their previous ones (Nat Comm 10.1038/ncomms5123) should be mentioned in the manuscript, why the Fe-NC could be better than the Fe(OAc)₂/C (Table 1 of the mentioned manuscript)?

Reviewer #2 (Remarks to the Author):

The authors developed MOF-derived iron-single site catalysts on nitrogen-doped carbon for ammoxidation of alcohols under ambient conditions. The catalytic performance is excellent compared with that of reported similar iron catalyst systems, which often require high temperature and pressure. Remarkably, the system is applicable for aliphatic substrates. There is enough novelty and inventiveness in their results. On the other hand, the effect of benzylamine additive during the catalyst preparation is unclear and an addition of more scientific discussion would be needed. In conclusion, I recommend to publish this paper in Nature Commun. after revisions suggested below.

1) I could not understand what is the motivation to add benzylamine during MOF construction. They did not provide any related literature and hypothesis. Why did the authors think that the additive could tune morphology in water? There is nothing convincing. Is there any literature demonstrating that a basic additive change the morphology of MOF? If so, the authors should cite appropriate literature and explain the logic of their strategy. If benzylamine simply worked as a base to accelerate the coordination of imidazole to metals by deprotonation, how about other basic additives? I imagined that the authors tried other basic additives and/or their equivalents. If they have such optimization data, it is very helpful for readers to provide them.

Reviewer #3 (Remarks to the Author):

This manuscript presents interesting results using modified ZIF-based Fe-N-C catalysts to the oxidative production of nitriles from alcohols. It builds on recent work of the investigators developing ZIF-based catalysts as attractive alternatives to noble-metal based catalysts, and follows on earlier work by them in synthesizing a broad range of nitriles via ammoxidation over earlier-generation catalysts. While well done, it is not of sufficiently broad interest to appeal to the general readership of Nature Communications. It would be a very suitable contribution to another journal more focused on pharmaceutical chemical reactions, or on design of advanced catalytic materials.

The new catalyst materials do provide an advance in selective oxidative amination of alcohols to produce a wide range of nitriles. They are highly active for the standard benzyl alcohol substrate, and are demonstrated for a wide range of interesting substrates with different functionalizations. An extensive range of nitriles are known to be of interest to the pharmaceutical chemistry community – that portion of the manuscript truly does educate the general reader but does not add to the impact on the most interested colleagues. The authors make claims for improved reactivity and mild conditions, but many substrates require stronger forcing conditions as described in the manuscript and the SI. More comparison with prior art and quantitative treatment are needed to support these statements if they are the central advance of the study.

The modification of the ZIF synthesis through addition of large amounts of benzylamine is very intriguing, and the impact on the as-synthesized crystallite habit and the activated catalyst reactivity is notable. This study reports notably high activity for a single, specific catalyst formulation, which is quite at odds with the statement in the Abstract about “tuning” the catalyst to achieve the “optimal catalyst”. Such claims would require an exploration of the ZIF-Bn and Fe-ZIF-Bn synthesis conditions, such as the ratio of Bn to 2-Melm, pyrolysis conditions, and other options. The 4:4 ratio of Bn:2-Melm is intriguing, and the excess over the Zn in the synthesis conditions opens many interesting questions about the actual incorporation in the solid and the effects of this interloper on the ZIF structure and its apparent reduction in defects and improvement in crystal habit. Apparently this affects the dispersion of Fe and the resistance to aggregation – very important issues in this arena. A more thorough study would be very eagerly received by the ZIF/MOF community in context of other investigations in that materials field. The authors may have already explored some range of synthesis conditions and be developing insights into how the Bn alters the several properties alluded to. Regardless, the results reported here cannot be viewed as presenting an isolated, unique catalyst formulation, but rather should be understood as demonstrating a useful additional synthetic factor for generating these intriguing ZIF-based catalyst precursors with improved metal-supporting capabilities.

The emphasis inserted on the green nature of the new catalyst synthesis compared to earlier methods based on organic solvents seems a side issue compared to the much larger green impact of a durable catalyst throughout its operating lifetime. This curiosity could be relegated to the SI rather than

competing with the more notable chemical impacts that do add to the impact of this advance in catalyst activity and selectivity. The extensive catalyst characterization is thorough and convincing in establishing the high dispersion of Fe sites in accord with other recent papers that focus on synthesis and characterization of single-atom catalysts. However, that is not the advance of this work.

To conclude, this work could be better presented to prominent journals in more specialized communities such as organic/pharmaceutical chemistry or materials synthesis/design. I do not recommend it for publication in Nature Communications.

Reviewer(s)' Comments to Author:

Reviewer: 1

Sun et al. reported the ammoxidation of alcohols to nitriles using the Fe-N on carbon materials derived from ZIFs. Compared with the previous work, a significant breakthrough is the catalyst could work mild reaction temperatures and pressures. The results are interesting and might be suitable for publication in Nature Communications. However, how could the catalyst work to outperform the previous ones is still not fully discussed. I recommended publication of the manuscript after a revision.

Answer: We thank the reviewer for the comments and suggestion. The reasons for the improved performance of the catalyst are discussed in more detail in the revised manuscript (see below).

(1) The reaction process involving multiple reaction steps should be discussed. Generally, the dehydrogenation of alcohol to aldehyde/ketone is a rate-control step (RDS). How about the RDS of this Fe-NC catalyzed reaction? Will the Fe-NC could catalyze the alcohol dehydrogenation under the given reaction conditions?

Answer: Based on a series of control experiments and further literature studies, the well-known reaction pathway for the synthesis of nitriles from alcohols is shown in Fig. S10. In this reaction sequence, the catalytic oxidation of benzyl alcohol to benzaldehyde is the rate-limiting step. Relevant discussion has been added to the manuscript.

(2) Attention should be focused on the stability of such Fe-NC catalyst. In order to confirm its heterogeneous feature, the classical hot-filtration experiment and ICP analysis of the reaction liquor should be performed.

Answer: As requested we added the hot-filtration experiment (Fig. S11) and ICP-OES analysis (Table S6) of the recycled catalysts to SI. The corresponding discussion about this part has been added to the manuscript.

(3) Comparison of the Fe-NC catalyst with their previous ones (Nat Comm 10.1038/ncomms5123) should be mentioned in the manuscript, why the Fe1-NC could be better than the Fe(OAc)₂/C (Table 1 of the mentioned manuscript)?

Answer: Compared with Fe₂O₃-NGr/C (Nat Comm 10.1038/ncomms5123), the here presented Fe₁-N-C material has a more stable atomic dispersion of Fe-N₄ active sites, large surface area, higher pore volume, and increased N content, which all might contribute to the increased reactivity. As requested by the referee, the comparative data of catalysts Fe₁-N-C and Fe₂O₃-NGr/C (Table 1 and Table S2) and the relevant discussion has been added to the manuscript.

Reviewer: 2

The authors developed MOF-derived iron-single site catalysts on nitrogen-doped carbon for ammoxidation of alcohols under ambient conditions. The catalytic performance is excellent compared with that of reported similar iron catalyst systems, which often require high temperature and pressure. Remarkably, the system is applicable for aliphatic substrates. There is enough novelty and inventiveness in their results. On the other hand, the effect of benzylamine additive during the catalyst preparation is unclear and an addition of more scientific discussion would be needed. In conclusion, I recommend to publish this paper in Nature Commun. after revisions suggested below.

Answer: We thank the reviewer for the helpful comments and suggestions. We have addressed all these comments and revised the manuscript and Supporting Information document accordingly.

(1) I could not understand what is the motivation to add benzylamine during MOF construction. They did not provide any related literature and hypothesis. Why did the authors think that the additive could tune morphology in water? There is nothing convincing. Is there any literature demonstrating that a basic additive change the morphology of MOF? If so, the authors should cite appropriate literature and explain the logic of their strategy.

Answer: This so-called 'coordination modulation strategy' has been reported in the literature for controlling the MOF particle size, morphology, and nucleation rate. Specifically, the use of cetyltrimethylammonium bromide (CTAB) as a surfactant or different organic amines (such as amino acids, triethylamine and n-butylamine) is known to control the particle morphology to a certain extent. Benzylamine was used by us as a crystallization regulator to compete with bridging ligands and therefore to regulate the crystallization process. The relevant literature regarding this strategy is cited as references 54-57. In addition, we added some more discussion about this part into the manuscript.

(2) If benzylamine simply worked as a base to accelerate the coordination of imidazole to metals by deprotonation, how about other basic additives? I imagined that the authors tried other basic additives and/or their equivalents. If they have such optimization data, it is very helpful for readers to provide them.

Answer: Several Fe-N-C nanocomposites derived from Fe-ZIFs particles with different ratios of benzylamine/2-Melm and different basic additives including organic amines (triethylamine) and inorganic bases (NaOH) were synthesized and added to Table S3. Compared to other additives benzylamine is the "best" crystallization regulator and directing agent for the synthesis of Fe-ZIFs. Further discussion about this part has been added to the manuscript.

Reviewer: 3

This manuscript presents interesting results using modified ZIF-based Fe-N-C catalysts to the oxidative production of nitriles from alcohols. It builds on recent work of the investigators developing ZIF-based

catalysts as attractive alternatives to noble-metal based catalysts, and follows on earlier work by them in synthesizing a broad range of nitriles via ammoxidation over earlier-generation catalysts. While well done, it is not of sufficiently broad interest to appeal to the general readership of Nature Communications. It would be a very suitable contribution to another journal more focused on pharmaceutical chemical reactions, or on design of advanced catalytic materials.

Answer: We thank the reviewer for his/her comments and suggestions. We agree that this work (like most scientific works) builds on recent works of the two cooperation partners. However, we believe by combining the expertise of the partners in materials synthesis and oxidation catalysis a significantly improved catalyst system has been developed, which shows unprecedented reactivity for such transformations in the presence of iron-based catalysts. Therefore, we believe publication is justified in Nature Communications (and two other referees share this opinion).

The new catalyst materials do provide an advance in selective oxidative amination of alcohols to produce a wide range of nitriles. They are highly active for the standard benzyl alcohol substrate, and are demonstrated for a wide range of interesting substrates with different functionalizations. An extensive range of nitriles are known to be of interest to the pharmaceutical chemistry community-that portion of the manuscript truly does educate the general reader but does not add to the impact on the most interested colleagues. The authors make claims for improved reactivity and mild conditions, but many substrates require stronger forcing conditions as described in the manuscript and the SI. More comparison with prior art and quantitative treatment are needed to support these statements if they are the central advance of the study.

Answer: We agree with the referee that functionalized nitriles are especially of interest to the pharmaceutical and agrochemical communities. Compared to other (non-)noble metal catalysts for ammoxidation processes, Fe1-N-C allows the oxidation of an array of diverse aryl, heterocyclic, and allylic alcohols under very mild conditions. However, for aliphatic alcohols, the reaction generally requires higher temperature and pressure. This is (and was) clearly stated in the manuscript. Nevertheless, the reactivity of the here presented catalyst also for aliphatic alcohols is still significantly better than other reported catalysts. To make this point clearer, comparative data of Fe1-N-C and other catalysts reported in the literature have been added to the manuscript (Table S1 and Table S2).

The modification of the ZIF synthesis through addition of large amounts of benzylamine is very intriguing, and the impact on the as-synthesized crystallite habit and the activated catalyst reactivity is notable. This study reports notably high activity for a single, specific catalyst formulation, which is quite at odds with the statement in the Abstract about "tuning" the catalyst to achieve the "optimal catalyst". Such claims would require an exploration of the ZIF-Bn and Fe-ZIF-Bn synthesis conditions, such as the ratio of Bn to 2-Melm, pyrolysis conditions, and other options. The 4:4 ratio of Bn:2-Melm is intriguing, and the excess over the Zn in the synthesis conditions opens many interesting questions about the actual incorporation in the solid and the effects of this interloper on the ZIF structure and its apparent reduction in defects and improvement in crystal habit. Apparently this affects the dispersion of Fe and the

resistance to aggregation – very important issues in this arena. A more thorough study would be very eagerly received by the ZIF/MOF community in context of other investigations in that materials field. The authors may have already explored some range of synthesis conditions and be developing insights into how the Bn alters the several properties alluded to. Regardless, the results reported here cannot be viewed as presenting an isolated, unique catalyst formulation, but rather should be understood as demonstrating a useful additional synthetic factor for generating these intriguing ZIF-based catalyst precursors with improved metal-supporting capabilities.

Answer: We thank the referee for these comments. We agree that the statement in the abstract can be misleading and therefore changed it. As mentioned by this referee before, the main emphasis of this work is the development of a better ammoxidation methodology. Consequently, aspects for the material community are less pronounced discussed. Nevertheless, several Fe-N-C nanocomposites derived from Fe-ZIFs particles with different ratios of benzylamine/2-Melm and different organic and inorganic additives such as triethylamine and NaOH were synthesized and added to Table S3. In addition, zinc was introduced as a self-sacrificial template to eliminate the aggregation of iron nanoparticles and improve the specific surface areas of Fe¹-N-C during the pyrolysis of Fe-ZIF-Bn. The relevant discussion about these points has been added to the manuscript.

The emphasis inserted on the green nature of the new catalyst synthesis compared to earlier methods based on organic solvents seems a side issue compared to the much larger green impact of a durable catalyst throughout its operating lifetime. This curiosity could be relegated to the SI rather than competing with the more notable chemical impacts that do add to the impact of this advance in catalyst activity and selectivity. The extensive catalyst characterization is thorough and convincing in establishing the high dispersion of Fe sites in accord with other recent papers that focus on synthesis and characterization of single-atom catalysts. However, that is not the advance of this work.

Answer: As the main aspect of this works is the development of a “greener” catalyst for general ammoxidations, we think the corresponding parts should stay in the manuscript and should not shifted to the SI.

Reviewer #1 (Remarks to the Author):

The comments have been well responded, and modification to the manuscript has been made accordingly, I suggest publication of it in nature communications.

Reviewer #2 (Remarks to the Author):

The authors address all the issues that this reviewer pointed out. Thus, I recommend to accept this paper after the following minor revision.

1. Please provide the yield of ZIF-8 synthesis and the catalyst synthesis in the Methods section.

Reviewer(s)' Comments to Author:

Reviewer: 1

The comments have been well responded, and modification to the manuscript has been made accordingly, I suggest publication of it in nature communications.

Answer: Thank you.

Reviewer: 2

The authors address all the issues that this reviewer pointed out. Thus, I recommend to accept this paper after the following minor revision.

(1) Please provide the yield of ZIF-8 synthesis and the catalyst synthesis in the Methods section.

Answer: As requested, we added the yield of ZIF-8 synthesis and the catalyst synthesis to the "Methods section".